# Circadian Light Manipulation and Melatonin Supplementation Enhance Morphine Antinociception in a Neuropathic Pain Rat Model

**DOI:** 10.3390/ijms26157372

**Published:** 2025-07-30

**Authors:** Nian-Cih Huang, Chih-Shung Wong

**Affiliations:** 1Department of Anesthesiology, Tri-Service General Hospital, Taipei 114, Taiwan; niancih@hotmail.com; 2Graduate Institute of Medical Sciences, National Defense Medical University, Taipei 114, Taiwan; 3Department of Anesthesiology, Cathay General Hospital, #280, Section 4, Renai Road, Taipei 106, Taiwan

**Keywords:** melatonin, morphine tolerance, neuropathic pain, circadian rhythm, neuroinflammation, astrocytes, light exposure, opioid-sparing therapy

## Abstract

Disruption of circadian rhythms by abnormal light exposure and reduced melatonin secretion has been linked to heightened pain sensitivity and opioid tolerance. This study evaluated how environmental light manipulation and exogenous melatonin supplementation influence pain perception and morphine tolerance in a rat model of neuropathic pain induced by partial sciatic nerve transection (PSNT). Rats were exposed to constant darkness, constant light, or a 12 h/12 h light–dark cycle for one week before PSNT surgery. Behavioral assays and continuous intrathecal (i.t.) infusion of morphine, melatonin, or their combination were conducted over a 7-day period beginning immediately after PSNT. On Day 7, after discontinued drugs infusion, an acute intrathecal morphine challenge (15 µg, i.t.) was administered to assess tolerance expression. Constant light suppressed melatonin levels, exacerbated pain behaviors, and accelerated morphine tolerance. In contrast, circadian-aligned lighting preserved melatonin rhythms and mitigated these effects. Melatonin co-infusion attenuated morphine tolerance and enhanced morphine analgesia. Reduced pro-inflammatory cytokine expression and increase anti-inflammatory cytokine IL-10 level and suppressed astrocyte activation were also observed by melatonin co-infusion during morphine tolerance induction. These findings highlight the potential of melatonin and circadian regulation in improving opioid efficacy and reduced morphine tolerance in managing neuropathic pain.

## 1. Introduction

Managing chronic pain requires a multifaceted approach aimed at reducing pain intensity, restoring function, and improving overall quality of life [1]. Although opioids are commonly used for pain management, their long-term application is limited by adverse effects and the development of tolerance. To enhance opioid efficacy while minimizing side effects, adjunctive strategies targeting endogenous regulatory systems have gained increasing interest.

Circadian rhythm disruption has been implicated in heightened pain sensitivity and opioid tolerance, particularly in chronic neuropathic pain [2,3]. Aligning analgesic interventions with the body’s intrinsic biological rhythms may improve both pain control and sleep quality [4,5]. Melatonin, an endogenous hormone secreted by the pineal gland, plays a central role in regulating circadian rhythms and has demonstrated anti-nociceptive, anti-inflammatory, and opioid-sparing properties in preclinical models [6,7]. In clinical settings, patients with chronic neuropathic pain often suffer from poor sleep quality and circadian misalignment, which further exacerbate pain perception and reduce opioid effectiveness [8]. This interplay underscores the rationale for targeting both circadian and opioid pathways to improve therapeutic outcomes. Exogenous melatonin supplementation has been shown to enhance opioid analgesia and attenuate the development of tolerance without inducing direct analgesic effects [9,10]. These findings support the potential use of melatonin as a co-therapeutic agent in chronic pain management, particularly under conditions where circadian misalignment exacerbates pain behaviors and impairs opioid efficacy [11,12].

In this study, we investigated how circadian light manipulation and exogenous melatonin supplementation modulate pain sensitivity and morphine tolerance in a rat model of neuropathic pain induced by partial sciatic nerve transection (PSNT). By comparing environmental light cues with pharmacological melatonin replacement, we aimed to explore their respective and combined roles in preserving opioid analgesia and mitigating neuroinflammation.

## 2. Results

### 2.1. Circadian Regulation of Serum Melatonin Under Light Exposure and Treatment Conditions

To evaluate the endogenous circadian pattern of melatonin secretion, serum melatonin levels were measured every hour from 10 p.m. to 10 a.m. on Day 4 (prior to surgery). As shown in Figure 1A, all three groups—sham, PSNT, and PSNT + morphine (n = 5 per group)—exhibited a typical nocturnal melatonin rhythm, characterized by a gradual rise beginning at midnight and peaking between 2 a.m. and 4 a.m. Peak melatonin concentrations ranged from 75 to 80 pg/mL, confirming the preservation of an intact circadian melatonin rhythm across all groups prior to any surgical or pharmacological intervention.

To assess the statistical significance of melatonin fluctuations across experimental groups during the dark phase, one-way ANOVA was performed at each hourly time point from 10 p.m. to 6 a.m. As presented in Table 1, significant group differences were observed at all time points, with particularly strong effects detected at 1 a.m. (F(3,16) = 48.147, *p* < 0.0001), 3 a.m. (F(3,16) = 36.596, *p* < 0.0001), and 4 a.m. (F(3,16) = 32.430, *p* < 0.0001), aligning with the expected nocturnal peak in melatonin secretion.

To investigate how different light environments influence melatonin secretion after PSNT and drug administration, peak serum melatonin levels (2–4 a.m.) were measured on Day 7 under three lighting conditions: constant darkness (dark–dark, DD), a standard 12 h light–dark cycle (dark–light, DL), and constant light (light–light, LL). These values were compared to corresponding baseline levels in Figure 1A. In the sham group (Figure 1B), melatonin levels remained relatively stable under DD and DL conditions, with peak values of 41.00 ± 3.61 and 35.33 ± 3.06 pg/mL, respectively. However, constant light exposure (LL) significantly suppressed melatonin output, with a reduced peak of 19.33 ± 5.13 pg/mL. In the PSNT group (Figure 1C), nerve injury alone led to an overall decline in melatonin secretion; the highest melatonin level was observed under DD (38.67 ± 3.06 pg/mL), while LL exposure resulted in the lowest peak (17.67 ± 2.52 pg/mL), suggesting that both nerve injury and circadian disruption impair pineal melatonin production. In the PSNT + morphine group (Figure 1D), the suppression of melatonin secretion was more pronounced. Peak levels under DD and DL were 34.67 ± 3.06 and 31.67 ± 2.89 pg/mL, respectively, whereas LL exposure yielded a minimal peak of 17.67 ± 2.52 pg/mL. These findings indicate that morphine treatment in combination with light-induced circadian disruption has a cumulative negative impact on endogenous melatonin levels.

### 2.2. Effects of Light Exposure on Paw Withdrawal Threshold in PSNT Rats Treated with Morphine and Melatonin

To evaluate the influence of lighting conditions on the analgesic effects of morphine and the modulatory role of melatonin, we measured paw withdrawal thresholds across three different light exposure paradigms: constant darkness (dark–dark, 24 h), constant light (light–light, 24 h), and alternating light–dark cycles (dark–light, 12 h–12 h), as shown in Figure 2A–C. In all three lighting conditions, PSNT rats exhibited a marked reduction in paw withdrawal threshold compared to sham-operated controls, indicating developed neuropathic pain. In the dark–dark condition (Figure 2A), morphine administration improved mechanical threshold on Day 1, but this effect diminished rapidly, suggesting the development of tolerance. However, the co-infusion of melatonin significantly attenuated the decline of morphine’s analgesic efficacy. Similarly, under light–light exposure (Figure 2B), morphine-treated rats showed a sharp drop in withdrawal threshold after Day 1. In contrast, rats receiving both morphine and melatonin maintained a relatively higher threshold across the study period, although the rescue effect appeared less pronounced than in the dark–dark condition. Under circadian dark–light cycle (Figure 2C), the overall trends were comparable. The PSNT + morphine group again displayed rapid tolerance development, while co-administration with melatonin restored the threshold, peaking on Day 7. Together, these findings demonstrate that light exposure modulates both neuropathic pain sensitivity and morphine tolerance. Melatonin co-treatment effectively preserved morphine’s analgesic effect across all lighting conditions, particularly under dark–dark and dark–light environments.

### 2.3. Influence of Light Conditions on Weight-Bearing Test to PSNT and Morphine Treatments

To investigate how environmental lighting modulates the antinociceptive effects of morphine in neuropathic pain, weight-bearing force was measured daily over a 7-day period following partial sciatic nerve transection (PSNT) under three lighting conditions: constant darkness (dark–dark), constant light (light–light), and a 12 h light–dark cycle (dark–light), as shown in Figure 3A–C. This test reflects pain-associated mechanical imbalance, where increased force on the contralateral limb indicates discomfort in the injured paw. In all lighting environments, PSNT rats exhibited a marked increase in contralateral weight-bearing force compared to sham-operated controls, indicative of neuropathic pain. In the dark–dark condition (Figure 3A), morphine administration (15 μg/h) significantly reduced the force imbalance on day 1, but this effect diminished over time, consistent with the development of morphine tolerance. Co-administration of melatonin (3 μg/h) with morphine led to a sustained reduction in force values, with measurements returning toward baseline level by day 7, suggesting melatonin preserved analgesic efficacy. Under light–light exposure (Figure 3B), PSNT rats displayed a persistent weight-bearing imbalance, and morphine producing only partial and transient improvement. When melatonin co-infused, a modest but consistent improvement was observed; however, the effect was less robust than in the dark–dark condition. These findings suggest that constant light may interfere with both endogenous melatonin production and morphine responsiveness. In the dark–light cycle (Figure 3C), morphine initially reduced the force imbalance; however, the efficacy declined by day 3 and showed tolerance developed by morphine i.t. infusion. As expected, rats treated with both morphine and melatonin exhibited a pronounced and declined in contralateral force on the contralateral hindlimb, achieving a reversal of the imbalance among all groups by day 7. This indicates that melatonin may enhance morphine’s antinociceptive effect and potentially support better PSNT limb weight bearing under physiological circadian condition. Together, these results demonstrate that light exposure affects mechanical pain adaptation in PSNT rats and modulates the analgesic efficacy of morphine. Melatonin co-treatment attenuated the development of morphine tolerance and improved pain-related weight-bearing behavior, particularly under dark–dark and dark–light conditions. Constant light exposure showed the weakest response, reinforcing the importance of circadian integrity in pain modulation and opioid responsiveness.

### 2.4. Co-Infusion of Melatonin Attenuates Morphine Tolerance Development

To evaluate the effect of melatonin on morphine-induced analgesic tolerance, tail-flick responses were quantified as %MPE (Maximum Possible Effect) in rats receiving continuous intrathecal infusion of melatonin (3 µg/h) combined with either saline or morphine (15 µg/h) over a 7-day period. As shown in Figure 4A, %MPE on Day 1 was significantly elevated in both morphine-treated groups, reflecting a robust initial antinociceptive effect. By Day 7, the morphine + DMSO group exhibited a marked reduction in %MPE, indicative of tolerance development. In contrast, the morphine + melatonin group maintained significantly higher %MPE levels throughout the 7-day period, suggesting that melatonin co-infusion effectively delayed the progression of morphine tolerance (*p* < 0.05).

To further characterize the acute antinociceptive response on Day 7, an intrathecal morphine challenge (15 µg) was administered, and tail-flick latency was measured over time and expressed as %MPE (Figure 4B). The morphine + melatonin group displayed a pronounced peak in %MPE at 30 min post-injection, which remained elevated throughout the 120-min observation period, indicating preserved morphine efficacy. Conversely, the morphine + DMSO group exhibited a rapid decline in %MPE after the initial peak, consistent with the presence of tolerance. Notably, melatonin alone did not produce significant antinociceptive effects compared to the saline + DMSO group, confirming its lack of direct analgesic action. These findings demonstrate that co-administration of melatonin sustains morphine antinociception and mitigates tolerance development, supporting its potential as an opioid-sparing adjunct in neuropathic pain management.

### 2.5. Role of Inflammatory Cytokines

To evaluate the impact of light exposure and morphine administration on neuroinflammatory responses following partial sciatic nerve transection (PSNT), we quantified the expression levels of pro- and anti-inflammatory cytokines (TNF-α, IL-1β, and IL-10) in spinal cord tissues under three light conditions: continuous darkness (dark–dark, 24 h), continuous light (light–light, 24 h), and alternating light–dark cycle (dark–light, 12 h–12 h). As shown in Figure 5A, TNF-α levels were significantly elevated in the PSNT group compared to the sham group across all lighting conditions, indicating a robust inflammatory response to nerve injury. Morphine administration (15 μg/h) partially attenuated this increase, reducing TNF-α expression to intermediate levels between sham and PSNT groups. Similarly, Figure 5B demonstrates that IL-1β, another key pro-inflammatory cytokine, was upregulated following PSNT. This increase was evident under all lighting conditions and was moderately reduced by morphine treatment, suggesting an anti-inflammatory effect of morphine. In contrast, as illustrated in Figure 5C, the anti-inflammatory cytokine IL-10 was markedly suppressed in the PSNT group compared to sham animals. Notably, morphine co-treatment significantly enhanced IL-10 levels, restoring or even surpassing baseline expression. Among the lighting conditions, the dark–light (12 h–12 h) group showed aa elevation of IL-10 in the PSNT + morphine group, suggesting that a rhythmic light–dark cycle may potentiate morphine’s anti-inflammatory action. Taken together, these results indicate that PSNT induces a strong pro-inflammatory response characterized by increased TNF-α and IL-1β and decreased IL-10 expression. Morphine effectively reverses these trends, particularly enhancing IL-10 expression, with light exposure conditions exerting subtle modulatory effects on the cytokine profiles.

### 2.6. Melatonin Inhibited Astrocyte Activity in the Spinal Cord of Morphine-Tolerant Rats

To evaluate astrocytic responses under neuropathic and opioid treatment conditions, GFAP immunofluorescence staining was performed on spinal cord sections (Figure 6A). Minimal GFAP expression was observed in the sham group, while the PSNT (saline) group showed substantial astrocyte activation. Morphine administration resulted in a partial reduction in GFAP immunoreactivity. Notably, co-treatment with melatonin led to a further decrease in GFAP signal intensity. This observation was supported by the quantification of GFAP-positive cells (Figure 6B), which showed a significant reduction in the morphine + melatonin group compared to morphine alone. Collectively, these results indicate that melatonin exerts a dual effect in morphine-tolerant rats by enhancing antioxidant defense mechanisms and suppressing astrocyte reactivity. These effects may contribute to melatonin’s potential therapeutic role in modulating opioid-induced neuroinflammation and tolerance development.

## 3. Discussions

This study provides compelling evidence supporting the role of melatonin as an effective adjuvant in mitigating morphine tolerance and modulating neuropathic pain (NP) in a rat model of partial sciatic nerve transection (PSNT). Although opioids are typically less effective in neuropathic conditions compared to inflammatory pain states, opioids are still used as a major medication of multimodal pain management in clinical, the PSNT model was deliberately chosen due to its clinical relevance and stringent pharmacological profile [13]. Neuropathic pain is often resistant to conventional opioids, making it a suitable platform to evaluate both the development of opioid tolerance and the efficacy of adjunctive strategies [14]. For the anti-inflammatory and anioxidative properties of melatonin and poorly, co-infusion of melatonin with morphine significantly prolonged the analgesic effect, attenuated neuroinflammation, and preserved morphine’s antinociceptive efficacy across various lighting conditions, and improved sleep quality at night [15]. These findings underscore the multifaceted role of melatonin, not only in circadian regulation but also in pain modulation and neuroprotection, especially under neuropathic injury and chronic opioid exposure.

### 3.1. Influence of Light Conditions and Circadian Rhythms

Environmental light exposure emerged as a critical modulator of both pain sensitivity and morphine responsiveness in this neuropathic pain model. Continuous light exposure (light–light, 24 h), which induces circadian disruption and mimics sleep deprivation, significantly exacerbated pain-related behaviors and accelerated the development of morphine tolerance. These findings are consistent with previous reports indicating that disrupted circadian rhythms impair nociceptive processing and reduce opioid analgesic efficacy [16].

In contrast, the standard 12 h:12 h light–dark cycle (dark–light) preserved circadian rhythmicity and maintained morphine-induced analgesia, highlighting the therapeutic relevance of aligned light–dark patterns in chronic pain management [17]. Although constant darkness (dark–dark) resulted in the highest serum melatonin levels and sustained analgesic responses, prolonged exposure to continuous darkness may eventually desynchronize circadian rhythms and adversely affect mood, behavior, and metabolism [18].

Interestingly, while endogenous melatonin levels under circadian-preserving conditions were biologically rhythmic, they remained substantially lower than pharmacological levels achieved through exogenous administration. This suggests that physiological melatonin alone may be insufficient to elicit clinically meaningful effects in the context of neuropathic pain and opioid tolerance.

It is important to note that this study focused on melatonin’s opioid-sparing effects; however, the absence of melatonin-only treatment groups under each lighting condition limits our ability to assess melatonin’s independent analgesic or anti-inflammatory properties. As such, it remains unclear whether the observed effects are purely synergistic with morphine or partially independent. This limitation has been addressed in the Discussion, and we recommend future studies include melatonin-only groups under all light paradigms (LL, DL, and DD) to better characterize its standalone effects.

Furthermore, although lighting conditions were used to infer circadian modulation, we did not directly monitor sleep–wake cycles or locomotor activity. Given the central role of circadian regulation in our hypothesis, the lack of behavioral confirmation represents another limitation. Future investigations should integrate objective measures such as actigraphy or EEG-based sleep monitoring to validate the functional impact of light-induced circadian alterations.

### 3.2. Melatonin’s Role in Preventing Morphine Tolerance

Continuous intrathecal administration of melatonin (3 µg/h) markedly delayed the development of morphine tolerance over a 7-day treatment course. This dose was selected based on our pilot dose–response experiments and previous studies demonstrating that low-dose melatonin, while lacking direct antinociceptive effects, exerts significant modulatory influence when co-administered with opioids [15]. Additionally, the chosen morphine dose (15 µg/h) has been widely used in rodent neuropathic pain models to induce robust analgesia followed by tolerance, allowing the evaluation of tolerance-sparing interventions [19].

Importantly, the melatonin concentration employed here is supraphysiological—substantially exceeding the peak endogenous serum melatonin levels observed in our circadian rhythm studies (~75–80 pg/mL; see Figure 1A)—yet remains within a non-toxic pharmacological range, as supported by prior reports on melatonin dosing safety and efficacy in neuropharmacological models [10]. This approach was intentional to mimic the therapeutic levels necessary for modulating central neuroinflammatory and circadian pathways implicated in morphine tolerance.

Consistent with previous findings, our results show that melatonin co-administration enhanced morphine’s analgesic duration and mitigated tolerance, without producing independent antinociceptive effects. This supports the role of melatonin as a synergistic and tolerance-sparing adjunct, rather than a direct analgesic substitute [15,20]. Potential mechanisms underlying this effect may include attenuation of glial activation, regulation of redox balance, and modulation of circadian genes or opioid receptor signaling cascades [21].

### 3.3. Behavioral Assays and Functional Outcomes

Three behavioral assays—tail flick latency, mechanical paw withdrawal threshold, and weight-bearing force—were employed to evaluate nociceptive thresholds and neuropathic pain expression. Each test provided complementary insights into analgesic and functional outcomes. The tail flick test reflects spinal reflex-based nociception as clinical acute pain, whereas the paw withdrawal and weight-bearing tests assess mechanical hypersensitivity and postural compensation as clinical neuropathic pain symptoms, respectively [22,23].

Across all behavioral paradigms, morphine initially produced robust analgesic effects that gradually declined, indicating the onset of opioid tolerance. Co-administration of melatonin consistently attenuated this decline, preserving morphine-induced analgesia and improving behavioral outcomes, particularly under circadian-aligned lighting conditions (dark–dark and dark–light).

Notably, the improvement in weight-bearing symmetry observed in melatonin co-treated animals may not solely reflect analgesia but could also involve enhanced motor coordination or systemic effects. We acknowledge this limitation and recommend that future studies include additional assessments to distinguish sensory from motor contributions to postural recovery.

We also recognize that the relatively small sample sizes (n = 3–6 per group) may limit the statistical power of some comparisons. This limitation arose from technical constraints and post-surgical attrition. However, key experimental groups were maintained at n ≥ 5 to preserve analytical validity. Future studies with larger, more uniform cohorts are warranted to further validate these findings and strengthen reproducibility.

### 3.4. Cytokine Modulation: Inflammation and Analgesia

Pro-inflammatory cytokines, including TNF-α and IL-1β, are key mediators of neuropathic pain and contribute to central sensitization following nerve injury [24]. In this study, PSNT robustly increased spinal TNF-α and IL-1β expression under all lighting conditions, indicating sustained neuroinflammatory activity [25]. Morphine treatment partially attenuated these cytokine elevations, suggesting modest anti-inflammatory effects consistent with its central analgesic action.

Importantly, environmental light exposure significantly modulated the inflammatory profile. Under constant light (LL) conditions, TNF-α and IL-1β levels remained elevated even with morphine administration, suggesting that circadian disruption impairs morphine’s anti-inflammatory efficacy. In contrast, animals maintained under a 12 h:12 h light–dark cycle (DL) exhibited greater suppression of pro-inflammatory cytokines and an increase in IL-10, an anti-inflammatory mediator, highlighting the immune-modulatory advantage of circadian alignment.

IL-10 expression, which was suppressed by PSNT across all light conditions, was differentially restored by morphine treatment. The DL group demonstrated the highest IL-10 levels post-treatment, whereas the LL group showed minimal recovery, reinforcing the hypothesis that circadian disruption blunts opioid-induced immunoregulation. Although our findings emphasize the role of light conditions in shaping cytokine dynamics, a notable limitation is the absence of data on microglial activation, which plays a central role in neuroimmune signaling and opioid tolerance. Future studies should include markers of microglial reactivity (e.g., Iba1) and assess whether melatonin modulates glial function through direct or indirect mechanisms. Elucidating melatonin’s impact on microglial signaling will be crucial to fully characterize its anti-inflammatory potential beyond cytokine regulation.

#### Astrocyte Activation and Glial Modulation

Partial sciatic nerve transection (PSNT) induced a significant upregulation of astrocyte activation in the spinal dorsal horn, as evidenced by increased GFAP immunoreactivity, a hallmark of neuroinflammation and central sensitization in neuropathic pain models [26,27]. Morphine monotherapy led to partial attenuation of GFAP expression, indicating modest glial suppression. In contrast, co-infusion of melatonin with morphine resulted in a more pronounced reduction in GFAP-positive cells, suggesting that melatonin enhances the gliostatic action of morphine.

These results are consistent with melatonin’s well-documented anti-inflammatory properties, particularly its ability to downregulate astrocytic activation and reduce the release of pro-inflammatory mediators such as TNF-α and IL-1β, as well as excitatory neurotransmitters like glutamate [28]. By mitigating astrocyte-mediated neuroinflammation, melatonin may help preserve opioid efficacy and delay the onset of morphine tolerance [20].

Although this study primarily focused on astrocytic responses, microglial activation is also known to contribute to opioid-induced neuroplasticity and tolerance. Future investigations incorporating microglial markers such as Iba1 will be necessary to fully elucidate the glial network involved in melatonin’s action [29]. Moreover, melatonin may also influence opioid signaling by modulating μ-opioid receptor (MOR) function. Chronic morphine administration is associated with MOR desensitization via β-arrestin recruitment and disrupted cAMP signaling [15]. Through MT1/MT2 receptor activation and G-protein-coupled signaling cascades, melatonin may stabilize MOR signaling and attenuate tolerance development [30].

In conclusion, our findings suggest that melatonin exerts a multifaceted protective role against morphine tolerance by suppressing astrocytic activation, modulating neuroinflammation, and stabilizing opioid receptor function. These data highlight its therapeutic potential as a non-opioid adjunct in chronic pain management. Nonetheless, the use of male-only subjects represents a limitation, and sex-specific responses warrant further investigation in future studies.

## 4. Methods and Materials

### 4.1. Employ an Animal Model to Disrupt Circadian Rhythms and Induce Partial Sciatic Nerve Transection

A total of 48 male Wistar rats (350–400 g) were used in this study. Animals were randomly assigned to experimental groups using a computer-generated randomization sequence. For intrathecal catheterization, rats were anesthetized with phenobarbital (65 mg/kg, i.p.) and housed under a 12 h/12 h light–dark cycle with unrestricted access to food and water. Animals exhibiting neurological deficits or surgical complications were excluded from analysis.

To induce neuropathic pain, partial sciatic nerve transection (PSNT) was performed on Day 4 under isoflurane anesthesia. A small incision was made over the thigh to expose the sciatic nerve, which was carefully separated from surrounding tissues. A consistent partial transection was then made using microsurgical instruments to ensure reproducibility across animals.

All experimental procedures were approved by the National Defense Medical Center Animal Care and Use Committee (IACUC-23-256) and conformed to the Guiding Principles in the Care and Use of Animals of the American Physiological Society. All behavioral assessments were conducted by investigators blinded to group allocation to minimize observer bias.

### 4.2. Nociception Assessment

The tail-flick test is a widely utilized method for assessing nociception in animals, particularly in rodents. This procedure involves exposing the animal’s tail to a radiant heat source, such as a focused hot water bath (typically approximately 52 °C, while the animal is restrained to minimize movement). The latency/reaction time, measured in seconds until the rat flicks its tail, serves as an indicator of its nociceptive threshold or pain sensitivity. Shorter reaction times indicate higher sensitivity to pain, whereas longer times suggest a greater pain threshold. This test is particularly beneficial for evaluating the analgesic effects of various substances, such as melatonin and morphine, by comparing reaction times before and after drug administration.

### 4.3. Intrathecal Drug Administration

Intrathecal (i.t.) catheterization and drug delivery were performed according to protocols adapted from our previous studies. Adult male Wistar rats (350–400 g) were anesthetized with Zoletil^®^ (50 mg/kg, intraperitoneally), and two polyethylene catheters (8.5 ± 0.5 cm) connected to 3.5 cm Silastic^®^ tubing were implanted through the atlantooccipital membrane, with the tips positioned at the lumbar enlargement (L1–L2). One catheter was anchored subcutaneously at the dorsal head region for potential morphine challenge, while the other was connected to a mini-osmotic pump (Alzet model 2001, DURECT Corporation, Cupertino, CA, USA) implanted subcutaneously for continuous drug infusion at 1 µL/h for 7 days. The pumps were filled with morphine sulfate (15 µg/h), melatonin (3 µg/h), or both drugs for co-treatment. Control animals received vehicle solutions (saline or DMSO) as appropriate. All pump implantations were performed immediately after partial sciatic nerve transection (PSNT) under isoflurane anesthesia. A single dose of cefazolin (100 mg/kg) was administered postoperatively to prevent infection. Catheter placement was confirmed postmortem, and animals showing postoperative neurological deficits were excluded from this study.

### 4.4. Morphine Tolerance Assessment

#### 4.4.1. Baseline Measurement

Baseline tail-flick latencies were assessed using the hot water immersion method. The distal one-third of the rat’s tail was immersed in a water bath maintained at 52 ± 0.5 °C, and the latency to withdrawal was recorded. This served as the reference for each animal’s thermal nociceptive threshold. The average baseline latency was 3 s, with a 10 s cut-off to prevent tissue injury.

#### 4.4.2. Continuous Morphine Administration

To induce morphine tolerance, drugs were delivered intrathecally via osmotic mini-pumps (Alzet 2001, DURECT Corporation, Cupertino, CA, USA; 1 μL/h) for 7 consecutive days. Under 2% isoflurane anesthesia, two PE-10 catheters (8.0 cm) were implanted via the atlanto-occipital membrane—one connected to the pump for infusion, the other reserved for drug challenge on Day 7.

#### 4.4.3. Drug Preparation and Delivery

Morphine sulfate (15 μg/h) and melatonin (3 μg/h) were co-administered through the same catheter in combined-treatment groups. Due to limited solubility, melatonin was dissolved in 5% DMSO, freshly prepared prior to implantation. Control groups receiving morphine alone were also infused with morphine in 5% DMSO, while vehicle groups received saline + 5% DMSO to account for any solvent effects.

### 4.5. Experimental Groups

Animals were randomly assigned to four treatment groups (n = 5–6 per group), each receiving continuous infusion at 1 μL/h:PSNT + saline + DMSO (5%),PSNT + melatonin (3 μg/h) + saline,PSNT + morphine (15 μg/h) + DMSO (5%), andPSNT + morphine (15 μg/h) + melatonin (3 μg/h).

All rats underwent partial sciatic nerve transection (PSNT) prior to infusion. Animals were housed individually under a 12 h/12 h light–dark cycle with ad libitum food and water. Those exhibiting neurological deficits or catheter-related complications were excluded.

### 4.6. Tail-Flick Latency and Tolerance Evaluation

Tail-flick latencies were monitored over 7 days to assess morphine tolerance. Measurements were taken at baseline and regular intervals using the same hot water immersion protocol. A gradual reduction in latency among morphine-treated animals indicated the development of tolerance.

To evaluate residual antinociceptive efficacy, an acute intrathecal morphine challenge (15 μg) was administered on Day 7 after stopping the continuous infusion. Tail-flick latencies were recorded at 0, 30, 60, 90, and 120 min post-injection. A blunted analgesic response was interpreted as evidence of established opioid tolerance.

### 4.7. Behavioral Assessment of Weight-Bearing and Mechanical Tactile Allodynia

Neuropathic pain was quantified using a weight-bearing incapacitance tester (Linton Instrumentation, Norfolk, UK). Rats were gently positioned in a restraining chamber, ensuring that each hind paw rested on an independent force plate. The force exerted by each limb (in grams) was measured over a 5 s interval, and the mean value from three consecutive trials was used for analysis. Behavioral assessments were conducted at baseline and on postoperative Days 1, 3, 5, and 7 to evaluate changes in nociceptive thresholds and functional asymmetry following PSNT.

Mechanical tactile allodynia was measured by the paw withdrawal sensitivity was evaluated in the left posterior paw with a Dynamic Plantar Aesthesiometer (Ugo Basile, Comerio, Italy). Rats were enclosed in plastic chambers mesh beneath. After 30 min of acclimatization, the withdrawal threshold was measured with a progressive increase in weight in the range of 1 to 50 g through a blunt metal rod (0.5 mm) facing the plantar region of the paw. The procedure was repeated thrice with 2 min intervals and averaged by setting 50 g as a cut-off threshold.

### 4.8. Astrocyte Activation, and Cytokine Measurement

Astrocytes and microglia are the principal glial cell types involved in central nervous system (CNS) inflammation. GFAP (Glial Fibrillary Acidic Protein) serves as a classical marker of astrocyte activation, indicating structural remodeling and reactivity in response to injury. To assess glial responses, lumbar spinal cord tissues were fixed, cryo-sectioned, and subjected to antigen retrieval and blocking. Sections were then incubated with primary antibodies against GFAP (for astrocytes), followed by fluorophore-conjugated secondary antibodies and nuclear counterstaining. Stained slides were examined under a fluorescence microscope to evaluate cell morphology and density changes, enabling quantification of astrocytic reactivity across treatment groups.

In parallel, enzyme-linked immunosorbent assays (ELISAs) were performed on spinal cord lysates to quantify cytokine levels, including TNF-α, IL-1β, and IL-10. All concentrations were normalized to total protein content and expressed as pg/100 μg protein. This combination of histological and biochemical approaches allowed comprehensive assessment of the neuroinflammatory profile induced by PSNT and modulated by pharmacological interventions.

### 4.9. Data Analysis

The data are presented as the mean ± SEM. Graphical representations and statistical analyses were performed using GraphPad Prism version 6.01 and Microsoft Excel. Raw RT-PCR data were analyzed on the Qiagen GeneGlobe web portal (http://www.qiagen.com/geneglobe, accessed on 23 July 2025), which calculated fold change/regulation using the 2^−ΔΔCT^ method. Statistical significance was assessed using two-way ANOVA, Tukey’s multiple comparisons test, Bonferroni’s multiple comparisons test, and Student’s *t*-test.

## 5. Conclusions

This study underscores the therapeutic potential of melatonin as an adjunctive strategy for mitigating morphine-induced tolerance and enhancing analgesic efficacy in a neuropathic pain model. The findings demonstrate that environmental lighting conditions significantly influence both pain sensitivity and the antinociceptive effect of opioids. Specifically, maintaining animals under standard light–dark cycles preserved circadian rhythm and improved morphine analgesic efficacy, whereas constant light exposure exacerbated tolerance development and pain behaviors. These results highlight the critical role of circadian regulation in pain modulation and opioid pharmacodynamics.

Moreover, exogenous melatonin administration (3 μg/h) effectively delayed the progression of morphine tolerance across all lighting conditions without exerting direct antinociceptive effects. This supports its role as a synergistic, tolerance-sparing agent rather than a standalone analgesic. At the molecular level, co-administration of melatonin reduced astrocyte activation, as evidenced by decreased GFAP expression, and modulated inflammatory cytokines by suppressing TNF-α and IL-1β while enhancing IL-10 levels.

Together, these findings offer compelling evidence that melatonin acts through both circadian and non-circadian pathways—modulating oxidative stress, neuroinflammation, and glial reactivity—to preserve opioid analgesia and prevent tolerance. The comprehensive behavioral and molecular analyses presented in this study provide a strong foundation for further investigation into melatonin’s mechanistic actions and support its potential clinical application in developing safer, more effective pain management strategies for patients requiring prolonged opioid therapy.

## Figures and Tables

**Figure 1 ijms-26-07372-f001:**
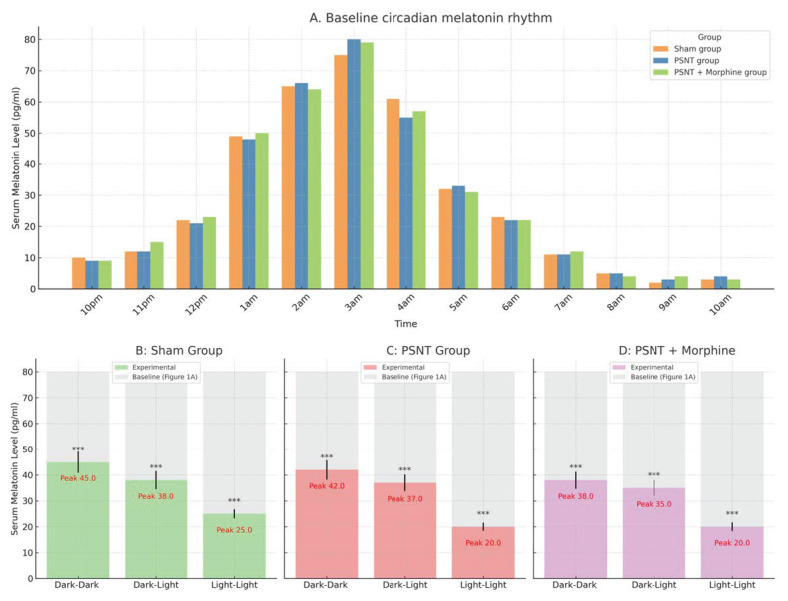
**Circadian variation in serum melatonin levels under different treatment and lighting conditions.** Serum melatonin concentrations were measured in rats subjected to various treatments and environmental light exposures. (**A**) Baseline circadian melatonin rhythm. Serum melatonin levels were assessed over a 12 h dark period (10 p.m. to 10 a.m.) on Day 4, prior to surgery. Three groups were compared: sham, PSNT, and PSNT + morphine (n = 5 per group). All groups exhibited a nocturnal surge in melatonin, peaking between 2–4 a.m. The overall pattern confirmed the presence of an intact circadian rhythm prior to intervention. Data are shown as mean ± SEM. (**B**–**D**) Melatonin peak levels under different lighting conditions. Post-surgical peak melatonin concentrations (Day 7) under three light exposure paradigms—constant darkness (dark–dark, DD), standard 12 h light–dark cycle (dark–light, DL), and continuous light (light–light, LL)—were compared against baseline values from (**A**) (light-colored bars). (**B**) Sham group: Peak melatonin levels were preserved under DD and DL conditions but markedly suppressed under LL exposure (Peak: 11.5 pg/mL), indicating circadian disruption by constant light. (**C**) PSNT group: Peak melatonin concentrations were moderately reduced across all light conditions, particularly under LL (Peak: 10.7 pg/mL), suggesting nerve injury impairs nocturnal melatonin output. (**D**) PSNT + morphine group: Melatonin levels were further diminished, especially under LL conditions (Peak: 10.1 pg/mL), indicating a combined suppressive effect of morphine and disrupted circadian cues. Bars represent the mean ± SEM; peak values are annotated in red. *** *p* < 0.001.

**Figure 2 ijms-26-07372-f002:**
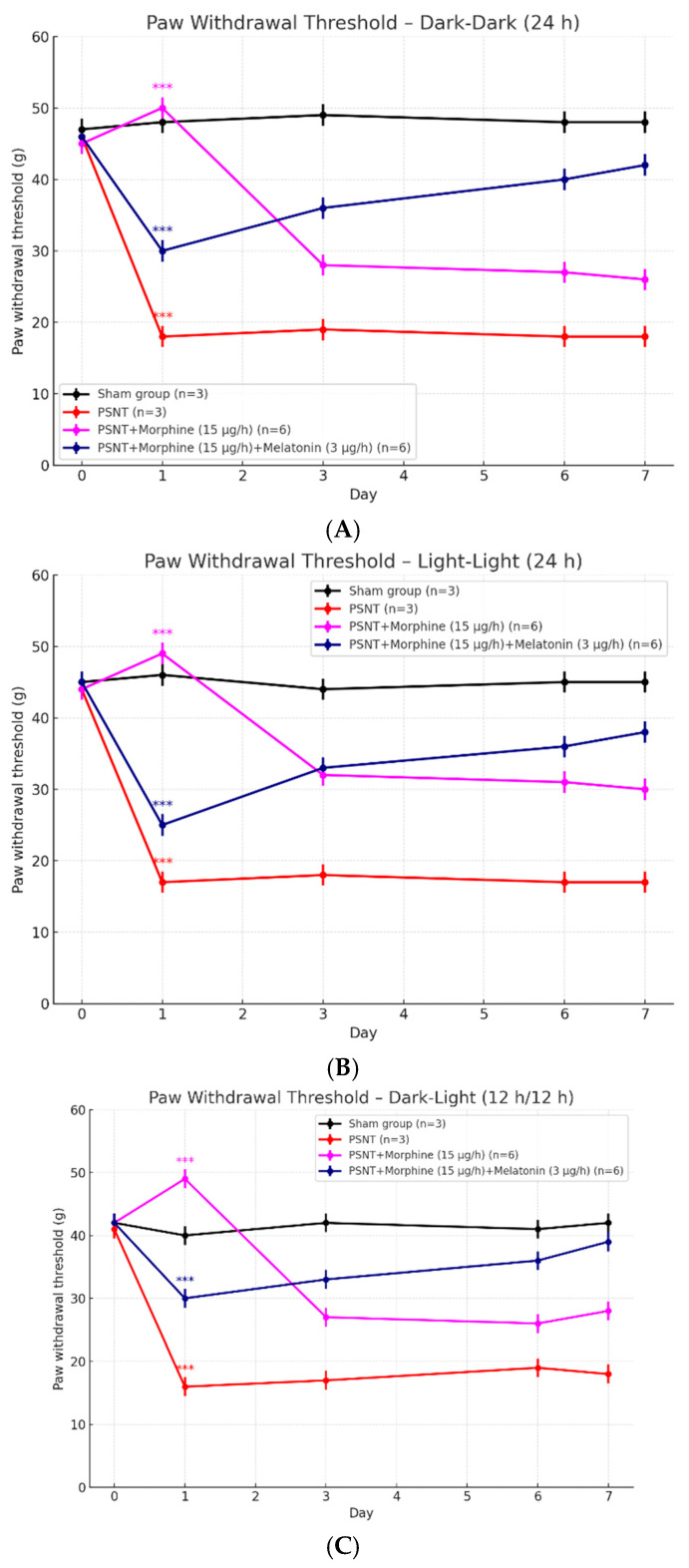
Effects of different light exposure conditions on paw withdrawal threshold for mechanical allodynia measurement in PSNT rats treated with morphine and melatonin. Line plots show the paw withdrawal threshold (g) measured over a 7-day period under three lighting conditions: (**A**) constant darkness (dark–dark, 24 h), (**B**) constant light (light–light, 24 h), and (**C**) standard 12 h light–dark cycle (dark–light, 12 h–12 h). Rats were divided into four groups: sham-operated controls (n = 3), PSNT (partial sciatic nerve transection, n = 3), PSNT + morphine (15 μg/h, n = 6), and PSNT + morphine + melatonin (3 μg/h, n = 6). Data are presented as the mean ± SEM. Statistical comparisons were performed against baseline (Day 0). *** *p* < 0.001. PSNT significantly reduced withdrawal thresholds under all lighting conditions. Morphine initially reversed this effect, but tolerance developed over time. Co-administration of melatonin preserved analgesic responses, especially under circadian-aligned lighting conditions (dark–dark and dark–light), indicating a modulatory role of melatonin in maintaining opioid efficacy.

**Figure 3 ijms-26-07372-f003:**
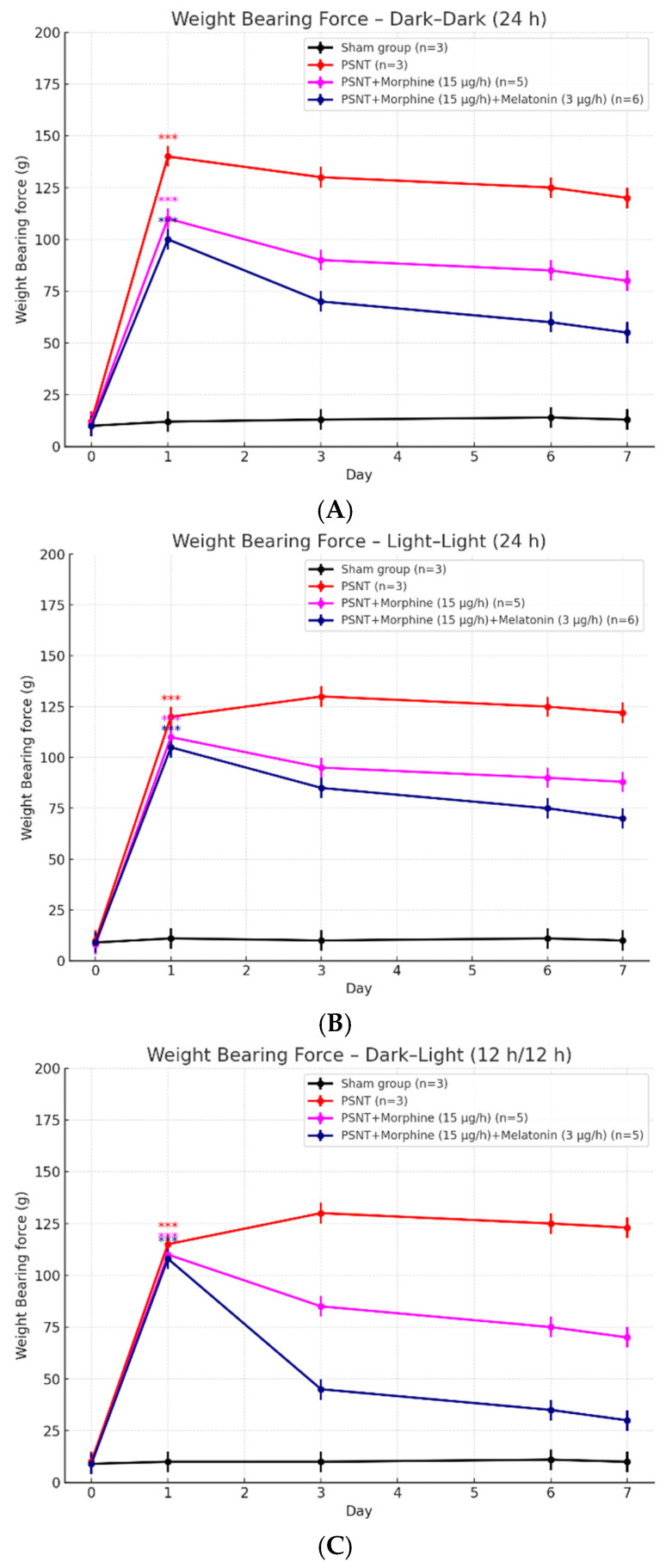
Effects of morphine and melatonin on weight-bearing asymmetry in PSNT rats under different lighting conditions. Line graphs illustrate daily changes in hind limb weight-bearing force (g) over a 7-day period following partial sciatic nerve transection (PSNT) in rats exposed to different light environments: (**A**) constant darkness (dark–dark, 24 h), (**B**) constant light (light–light, 24 h), and (**C**) standard 12 h–12 h light–dark cycle (dark–light). Experimental groups included sham controls (n = 3), PSNT (n = 3), PSNT + morphine (15 μg/h, n = 5), and PSNT + morphine + melatonin (3 μg/h, n = 5), with drugs delivered continuously via intrathecal infusion. All graphs show the mean ± SEM. Statistical comparisons were made against the baseline value (Day 0). *** *p* < 0.001. PSNT induced significant mechanical imbalance, characterized by increased contralateral weight-bearing. Morphine provided transient relief, but analgesic efficacy waned over time, indicating tolerance. Co-infusion of melatonin mitigated this effect and helped maintain weight-bearing symmetry, particularly under circadian-supportive conditions (dark–dark and dark–light).

**Figure 4 ijms-26-07372-f004:**
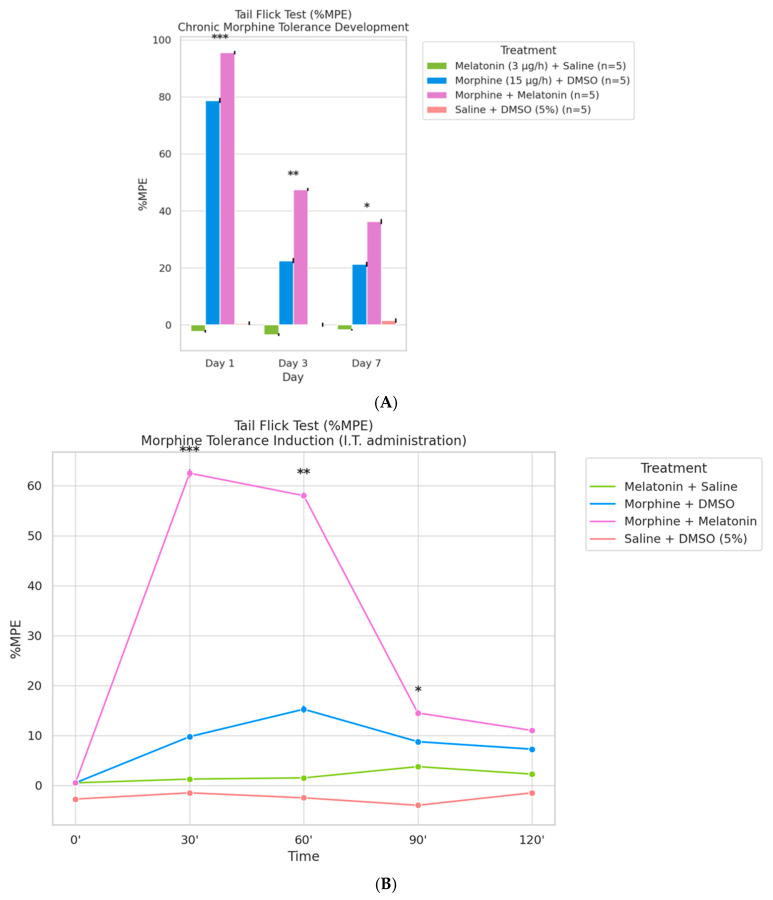
(**A**) Chronic morphine administration induces analgesic tolerance, which is attenuated by co-treatment with melatonin. Tail-flick latency was measured on Days 1, 3, and 7 following continuous intrathecal infusion of saline, morphine (15 µg/h), or morphine combined with melatonin (3 µg/h). Data are expressed as %MPE (Maximum Possible Effect), calculated using the formula: %MPE = [(post-drug latency − baseline latency)/(cut-off − baseline latency)] × 100, with a baseline latency of 2 s and a cut-off latency of 10 s. Bars represent the mean ± SEM (n = 5 rats per group). *p* < 0.001, *p* < 0.01, *p* < 0.05 compared to all other treatment groups at the same time point. (**B**) Melatonin enhances the acute antinociceptive effect of morphine and delays tolerance onset. Tail-flick latency was measured at 0, 30, 60, 90, and 120 min following a single intrathecal dose of morphine (15 µg/h), with or without melatonin (3 µg/h), in PSNT rats. Additional control groups received saline or melatonin alone. Data are presented as %MPE (Maximum Possible Effect), calculated using the formula: %MPE = [(post-drug latency − baseline latency)/(cut-off − baseline latency)] × 100, where baseline latency = 2 s and cut-off = 10 s. Each line represents the mean ± SEM (n = 5 per group). * *p* < 0.05, ** *p* < 0.01, and *** *p* < 0.001, for the morphine + melatonin group compared to all other groups at the corresponding time points (30′, 60′, and 90′).

**Figure 5 ijms-26-07372-f005:**
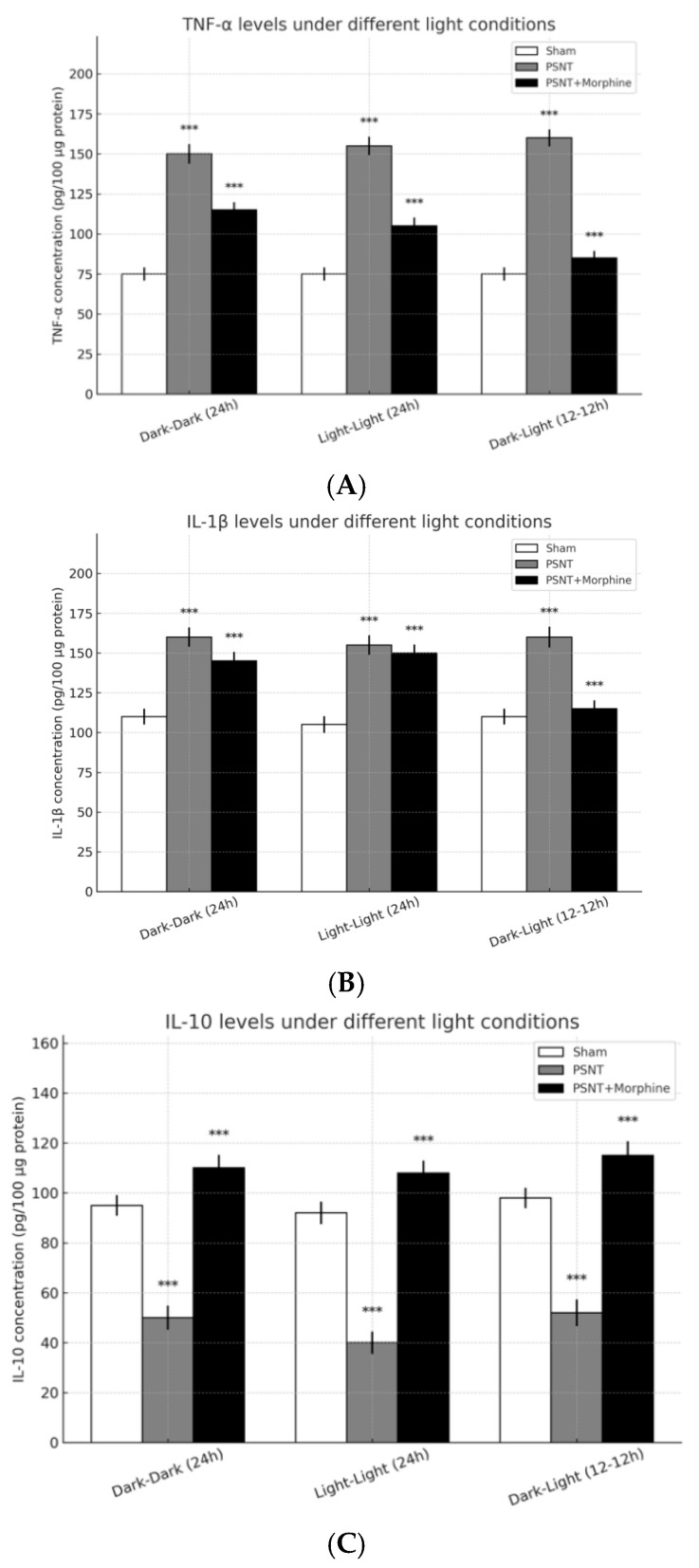
Effects of light exposure and morphine administration on inflammatory cytokine levels in spinal cord tissue following partial sciatic nerve transection (PSNT). Cytokine concentrations (pg/100 μg protein) were quantified in lumbar spinal cord samples from sham, PSNT, and PSNT + morphine groups under three light conditions: constant darkness (dark–dark, 24 h), constant light (light–light, 24 h), and standard 12 h–12 h light–dark cycle (dark–light). (**A**) TNF-α levels were significantly elevated in PSNT groups across all lighting conditions and partially reduced by morphine. (**B**) IL-1β levels mirrored TNF-α changes, with PSNT-induced increases attenuated to varying degrees by morphine treatment. (**C**) IL-10, an anti-inflammatory cytokine, was suppressed following PSNT but markedly upregulated in the PSNT + morphine group under all lighting conditions. Data are presented as the mean ± SEM (n = 5 per group). *** *p* < 0.001., relative to the sham group for each lighting condition unless otherwise indicated.

**Figure 6 ijms-26-07372-f006:**
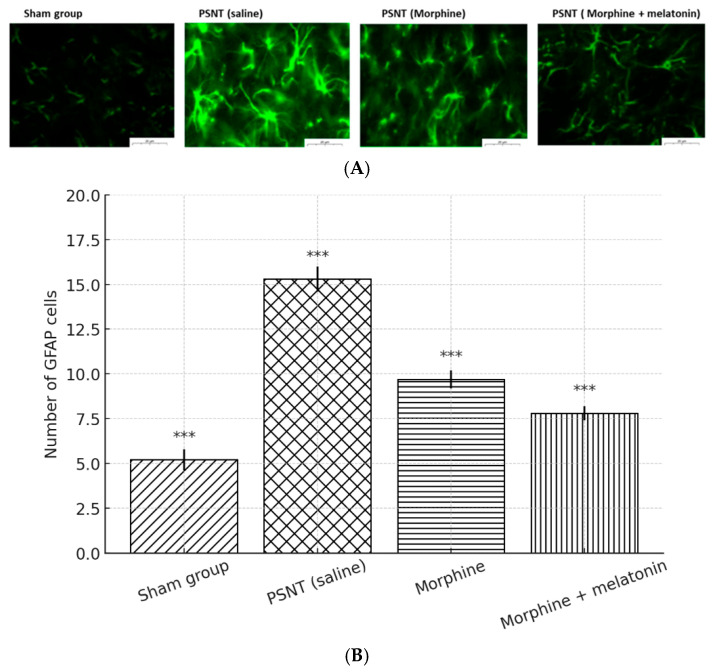
Melatonin attenuates PSNT- and morphine-induced astrocyte activation in the spinal cord. (**A**) Representative immunofluorescence images of GFAP-positive astrocytes in the dorsal horn of the lumbar spinal cord from four groups: sham, PSNT (saline), PSNT (morphine), and PSNT (morphine + melatonin). Astrocytes are stained green for GFAP. PSNT induced marked astrocytic activation, which was reduced by morphine and further suppressed by co-administration of melatonin. (**B**) Quantification of GFAP-positive cells per microscopic field. Data are presented as the mean ± SD (n = 5). PSNT significantly increased astrocyte numbers, while morphine and melatonin treatment progressively attenuated this elevation. *** *p* < 0.001, compared to the PSNT (saline) group unless otherwise indicated.

**Table 1 ijms-26-07372-t001:** **One-way ANOVA results of serum melatonin levels across treatment groups at hourly time points from 10 p.m. to 6 a.m.** F-values, degrees of freedom, and exact *p*-values are reported for each time point. Significant differences were observed at all time points, indicating treatment-related effects on melatonin levels during the dark phase. ANOVA was performed with 3 degrees of freedom between groups and 16 within groups.

Time	F-Value	*p*-Value	df_Between	df_Within
10:00 p.m.	9.725	0.0007	3	16
11:00 p.m.	16.296	0	3	16
12:00 a.m.	15.119	0.0001	3	16
1:00 a.m.	48.147	0	3	16
2:00 a.m.	6.067	0.0059	3	16
3:00 a.m.	36.596	0	3	16
4:00 a.m.	32.43	0	3	16
5:00 a.m.	13.263	0.0001	3	16
6:00 a.m.	5.377	0.0094	3	16

## Data Availability

The data supporting the findings of this study are available from the corresponding author upon reasonable request.

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
