# Peer review of "Circadian Light Manipulation and Melatonin Supplementation Enhance Morphine Antinociception in a Neuropathic Pain Rat Model"

_ijms, 2025, doi:10.3390/ijms26157372_

Round 1
Reviewer 1 Report
Comments and Suggestions for Authors
- Why did the authors choose a neuropathic pain model (PSNT), in which opioids are typically less effective compared to NSAIDs? Since neuropathic pain is often considered non-receptor-driven and opioids act via receptor mechanisms, this model might not fully reflect receptor-mediated analgesia.
- In subsection 2.2, I suggest the authors avoid general textbook-like descriptions of the tail-flick test. Instead, they should provide specific methodological parameters, such as water temperature and cut-off latency. Moreover, the test described corresponds more to a hot-water immersion tail-flick assay than to the classic radiant heat tail-flick method
- The manuscript states that morphine and melatonin were co-administered intrathecally. Could the authors clarify whether both drugs were delivered through the same catheter/pump system, and if so, how were they solubilized and combined? Information on solvents, compatibility, and formulation is essential for reproducibility
- I recommend that all pain-related behavioral data (tail-flick latency and others) be expressed as %MPE (Maximum Possible Effect), which is standard in opioid pharmacology and improves comparability across studies
- Please justify the selection of the intrathecal doses of morphine (15 µg/h) and melatonin (3 µg/h). Were these based on preliminary studies or prior literature? A dose-response analysis or citation would strengthen the rationale.
- DMSO (5%) was used in some treatment groups, but it is unclear which compounds were solubilized in DMSO and whether DMSO alone was tested for potential confounding effects. Please clarify which vehicles were used for each group and whether vehicle controls were appropriately implemented
- In Figures 2 and 3, the number of animals in sham and PSNT groups is only n=3. This small sample size may limit the statistical power and reliability of comparisons. Please justify this choice or provide power calculations.
- Was randomization used to assign animals to treatment groups? Were the experimenters blinded during behavioral assessments? These are important methodological details, especially in nociception studies where observer bias can influence outcomes.
English should be improved
Reviewer 2 Report
Comments and Suggestions for Authors
This manuscript presents a preclinical investigation into how circadian rhythm modulation can influence neuropathic pain behaviors and morphine tolerance in a PSNT rat model. The study integrates behavioral, biochemical, and histological assays to support the hypothesis that circadian alignment and melatonin co-treatment mitigate morphine tolerance and improve analgesic efficacy. Though the concept is scientifically interesting and translationally relevant, several methodological shortcomings, ambiguities in data presentation, and inconsistencies in interpretation must be addressed before the manuscript can be considered suitable for publication.
Here are my detailed comments and suggestions:
- The rationale behind dose selection for melatonin (3 µg/h) and morphine (15 µg/h) is not justified with previous literature or pilot dose-response experiments.
- Provide a scientific rationale for the dosing strategy based on prior pharmacological or pharmacokinetic studies.
- Discuss whether the selected dose of melatonin is physiologically or supraphysiologically relevant.
- Inclusion of a melatonin-only treatment group in light manipulation experiments (e.g., under LL and DL conditions) is suggested.
- Similarly inclusion of a PSNT + melatonin group under different light conditions is crucial to isolate the effect of melatonin alone on pain and cytokines.
- It appears there is a lack of control for sleep behavior or activity. Since circadian rhythm disruption is central to the hypothesis, sleep-wake patterns or locomotor activity should have been monitored.
- At a minimum, this limitation should be acknowledged, and future studies should consider related measurements to validate circadian disruption.
- The sample size (n = 3–6 per group) is low for behavioral assays and highly variable across experiments. This raises concerns about statistical power and reproducibility.
- Also the statistical reporting is incomplete. F-values, degrees of freedom, and exact p-values are not provided.
- It is unclear as the manuscript does not describe whether blinding and randomization were applied during experiments.
- The manuscript contains numerous grammatical errors, typographical mistakes, and awkward sentence structures that impair readability.
- Especially repetitive descriptions in the Introduction & Discussion sections dilute the scientific scope. Remove unnecessary repetitive statements of melatonin’s effects and streamline claims.
- Some figures lack the error bars, and significance indicators are not clearly defined.
- It appears that the interpretation of behavioral tests is overstated. For example, melatonin’s effect on weight-bearing is interpreted as analgesic, but this may also reflect improved motor function or systemic effects.
- In the cytokine assays, baseline cytokine levels from naive animals (without PSNT or drug treatment) are not presented, making it difficult to assess the magnitude of change.
- The authors propose anti-inflammatory and circadian mechanisms for melatonin’s effect but did not measure critical markers like microglial activity, clock gene expression, & melatonin receptor localization.
- Apparently, the role of microglia is ignored despite their well-established role in morphine tolerance. Discuss microglial involvement and suggest whether melatonin may also impact microglial activation.
- The exclusion of female rats limits the applicability of findings, particularly in pain research where sex differences are well-documented.
Round 2
Reviewer 1 Report
Comments and Suggestions for Authors
The Authors have now improved the manuscript, therefore I suggest its publication